

# Trends in Global Tropospheric Ozone Inferred from a Composite Record of TOMS/OMI/MLS/OMPS Satellite Measurements and the MERRA-2 GMI Simulation

Jerry R. Ziemke[1,2], Luke D. Oman[1], Sarah A. Strode[1,4], Anne R. Douglass[1], Mark A. Olsen[1,2], Richard D. McPeters[1], Pawan K. Bhartia[1], Lucien Froidevaux[3], Gordon J. Labow[5], Jacquie C. Witte[5], Anne M. Thompson[1], David P. Haffner[5], Natalya A. Kramarova[1], Stacey M. Frith[5], Liang-Kang Huang[5], Glen R. Jaross[1], Colin J. Seftor[5], Mathew T. Deland[5], Steven L. Taylor[5]

[1]NASA Goddard Space Flight Center, Greenbelt, Maryland, USA

[2]Morgan State University, Baltimore, Maryland, USA

[3]NASA Jet Propulsion Laboratory, California, USA

[4]Universities Space Research Association, Columbia, MD, USA

[5]SSAI, Lanham, Maryland, USA

**Abstract.**  Past studies have suggested that ozone in the troposphere has increased globally throughout much of the 20th century due to increases in anthropogenic emissions and transport. We show by combining satellite measurements with a chemical transport model that during the last four decades tropospheric ozone does indeed indicate increases that are global in nature, yet still highly regional. Satellite ozone measurements from Nimbus-7 and Earth Probe Total Ozone Mapping Spectrometer (TOMS) are merged with ozone measurements from Aura Ozone Monitoring Instrument/Microwave Limb Sounder (OMI/MLS) to determine trends in tropospheric ozone for 1979-2016. Both TOMS (1979-2005) and OMI/MLS (2005-2016) depict large increases in tropospheric ozone from the Near East to India/East Asia and further eastward over the Pacific Ocean. The 38-year merged satellite record shows total net change over this region of about +6 to +7 Dobson Units (DU) (i.e., ~15-20% of average background ozone), with the largest increase (~4 DU) occurring during the 2005-2016 Aura period. The Global Modeling Initiative (GMI) chemical transport model with time-varying emissions is included to evaluate tropospheric ozone trends for 1980-2016. The GMI simulation for the combined record also



depicts greatest increases of +6 to +7 DU over India/east Asia, identical to the satellite
measurements.  In regions of significant increases in TCO the trends are a factor of 2-2.5 larger
for the Aura record when compared to the earlier TOMS record; for India/east Asia the trends in
TCO for both GMI and satellite measurements are ~+3 DU-decade$^{-1}$ or greater during 2005-2016
compared to about +1.2 to +1.4 DU-decade$^{-1}$ for 1979-2016.  The GMI simulation and satellite
data also reveal a tropospheric ozone increase of ~+4 to +5 DU for the 38-year record over
central Africa and the tropical Atlantic Ocean. Both the GMI simulation and satellite-measured
tropospheric ozone during the latter Aura time period show increases of ~+3 DU-decade$^{-1}$ over
the NH Atlantic and NE Pacific.

**1. Introduction**

Over the last several decades there have been substantial regional changes in global pollutants
including precursors of tropospheric ozone as documented by many studies (e.g., Granier et al.,
2011; Parrish et al., 2013; Cooper et al., 2014; Lee et al., 2014; Zhang et al., 2016; Heue et al.,
2016; Lin et al., 2017).  The largest increases in global pollutants over the last four decades
occurred broadly over a region extending from the Near East to India and east/SE Asia.  Lin et
al. (2017) used a global chemistry-climate model (CCM) for 1980-2014 to study the effects of
global changes in emissions on surface ozone.  They show that rising increases in emissions,
including a tripling of Asian $NO_x$ (NO + $NO_2$) since just 1990, lead to large increases in surface
ozone over India/East Asia and to a lesser extent over the western US due to long-range
transport.   Shepherd et al. (2014, and references therein) suggest that increases in global
tropospheric ozone have occurred during much or most of the 20th century due to increases in
anthropogenic emissions.  The model simulation by Shepherd et al. (2014) indicates (their Figure
5) positive trends in global tropospheric ozone since 1960, primarily in the tropics and NH extra-
tropics.

The changes in global emissions since 1980 are described by Zhang et al. (2016) as an
equatorward redistribution over time into developing countries of India and those of SE Asia.
Zhang et al. (2016) used a global chemical-transport model (CTM) for 1980-2010 to quantify the
effects on tropospheric ozone from these changes in emissions.  The model simulations and



OMI/MLS satellite measurements employed by Zhang et al. (2016) indicated largest increases in
tropospheric ozone extending from the Near East to India and SE Asia and further eastward over
the Pacific Ocean. Zhang et al. (2016) included IAGOS aircraft ozone profiles that also showed
large increases (i.e., double-digit percent increases) for India, SE Asia, and East Asia between
the 1994-2004 and 2005-2014 time records. The model used by Zhang et al. (2016) also
simulated a net increase in global tropospheric ozone of about 28 Tg (~8.9%) over the 30-year
record. The results by Zhang et al. (2016) appear consistent with the Bulletin of the American
Meteorological Society BAMS State of the Climate Report for year 2016 that indicates about
21.8 Tg increase in OMI/MLS tropospheric ozone when averaged over 60ºS-60ºN between
October 2004 and December 2016, with largest contribution to global trends (about +3 to +4
DU-decade$^{-1}$ for OMI/MLS) originating from the same India and east/SE Asia region. The first
evidence of increases in tropospheric ozone over SE Asia from satellite data was shown by Beig
and Singh (2007). Beig and Singh used a version of Convective-Cloud Differential (CCD)
gridded tropospheric ozone for 1979-2005 that was a predecessor to the current CCD data used
for our study (discussed in Section 2). The CCD algorithm is described by Ziemke et al. (1998).
The largest increases in tropospheric ozone reported by Beig and Singh (2007) were up to 7-9%
decade$^{-1}$ and were located in SE Asia.

Gaudel et al. (2018) (i.e., Chapter 6 of the international Tropospheric Ozone Assessment Report,
TOAR) provides analyses of trends in tropospheric ozone calculated from a large array of data
sources including satellite, aircraft, balloon ozonesondes and surface measurements. Figure 24
of Gaudel et al., (2018) shows calculated linear trends/decadal changes during the Aura time
record for six global data products, five from satellite and one from trajectory-mapped
ozonesondes. The six products show large divergence in estimated trends, in part due to their
short and differing time records; it was noted that one should be careful about placing precise
numbers on estimated trends in TCO from the results. Figure 25 of Gaudel et al. (2018)
combined all six TCO products together statistically and showed that the largest and most
consistent (and positive) trends between the six products were centered over SE Asia.

Heue et al. (2016) derived a merged 1995-2015 tropical tropospheric ozone dataset from multiple
satellite instruments using a variant of the CCD approach for latitude range ±20º. Their dataset



was determined by concatenating measurements from several instruments including
SCIAMACHY and GOME (but not including either TOMS or OMI/MLS).  Their main findings
included evidence for increases in tropospheric ozone over both India/SE Asia and the tropical
Africa/Atlantic region; however, their largest detected positive trends were across tropical
Africa/Atlantic rather than India/SE Asia.  Heue et al. (2016) estimated a mean trend in TCO of
about +0.7 DU-decade$^{-1}$ in the tropics (15ºS-15ºN).  Leventidou et al. (2018) using similar (but
processed differently) SCIAMACHY/GOME CCD TCO measurements for 1995-2015 found
~+3 DU-decade$^{-1}$ trend over southern Africa, but no statistical change in the tropics (15ºS-15ºN).

The purpose of our study is to derive trends in tropospheric ozone for 1979-2016 by combining
TOMS (1979-2005) and OMI/MLS (2005-2016) measurements.  A main incentive is to evaluate
TCO trends for a longer satellite record than previous investigations including TOAR, and to
identify and possibly explain the regional trend patterns that emerge from the data.  Areal
coverage for calculated trends is all longitudes and latitudes 30ºS – 30ºN for TOMS and 60ºS-
60ºN for OMI/MLS.  The Global Modeling Initiative (GMI) chemical transport model (CTM)
replay simulation is included to assess ozone trends during both the TOMS and OMI/MLS time
periods.  All satellite ozone products were re-processed from previous versions to improve data
quality for trend calculations.  We also provide a preliminary evaluation of tropospheric column
ozone (TCO) measured from the Ozone Mapping Profiler Suite (OMPS) nadir-mapper and limb-
profiler instruments beginning in 2012 as possible future continuation of the OMI/MLS TCO
record.  Section 2 discusses the satellite measurements, GMI model, ozonesonde data, and trend
calculations.  Section 3 discusses derived trends in tropospheric ozone including net changes for
the combined 38-year record.  Results are summarized in Section 4.

**2. Satellite Measurements, MERRA-2 GMI Model, Ozonesondes, and Trend Calculations.**

2.1. Satellite Measurements.

All satellite measurements of TCO used for our study are developed within NASA Goddard
Code 614 and updated and upgraded periodically for the science community.    TCO
measurements and their validation from Nimbus-7 (N7) and Earth Probe (EP) TOMS



instruments are discussed by Ziemke et al. (2005, and references therein). TOMS TCO for 1979-
2005 is derived using the Convective-Cloud Differential (CCD) algorithm (Ziemke et al., 1998)
which differences clear versus thick cloud measurements of column ozone. Useful CCD gridded
TCO is limited mostly to tropical latitudes. Our TOMS CCD dataset originates from a
preliminary TOMS CCD gridded dataset that Beig and Singh (2007) used for evaluating TCO
trends, but now includes a re-processing with extensive flagging of outliers out to latitudes ±30°.
The N7 and EP TOMS instruments have similar spectral/spatial/temporal resolution with TCO
obtained from both using the same version 8 algorithm. TOMS TCO is determined by
subtracting thick cloud column ozone measurements (to estimate stratospheric column ozone,
SCO) from near clear-sky total column ozone. By differencing SCO and total ozone from the
same instrument, derived TCO is largely self-calibrating over time and should not be affected by
instrument/inter-instrument drifts or offsets. Standard precision error (i.e., 1σ standard
deviation) of TOMS gridded TCO is estimated to be about 1.7 DU (e.g., Ziemke et al., 1998).

We also include OMI/MLS TCO (Ziemke et al., 2006) for January 2005-December 2016 and
latitude range 60°S-60°N. TCO is determined by subtracting MLS SCO from OMI total column
ozone each day at each grid point. Tropopause pressure used to determine SCO invoked the
WMO 2K-km$^{-1}$ lapse-rate definition from NCEP re-analyses. For consistency these same lapse-
rate tropopause pressure fields were used to derive TCO for ozonesondes, OMPS, and the GMI
model (discussed below). OMI total column ozone is retrieved using the OMTO3 v8.5 algorithm
that includes in situ UV cloud pressures from OMI (Vasilkov et al., 2008) and several other
improvements from version 8. The OMI total ozone and cloud data including discussion of data
quality are available from https://ozoneaq.gsfc.nasa.gov/. The MLS data used to obtain SCO
were derived from their v4.2 ozone profiles (https://mls.jpl.nasa.gov/data/datadocs.php/). We
estimate 1σ precision for the OMI/MLS monthly-mean gridded TCO product to be about 1.3
DU. The additional Supporting Material discusses both validation and adjustments made to
OMI/MLS TCO. It can be shown that OMI/MLS TCO derived from this residual technique is
nearly identical to the TCO from OMI CCD measurements for the same time period, albeit with
the CCD data limited mostly to tropical/subtropical latitudes (e.g., Ziemke et al., 2012).



Tropospheric ozone for January 2012 through 2016 is also determined from the OMPS nadir-
mapper and limb-profiler instruments onboard the National Polar-orbiting Operational
Environmental Satellite System (NPP) spacecraft. The OMPS tropospheric ozone is evaluated
for possibly continuing the OMI/MLS data record. TCO is determined by subtracting OMPS
v2.5 limb-profiler SCO from OMPS v2.3 nadir-mapper total column ozone. SCO is determined
from the limb-profiler measurements using the same tropopause pressure fields as for MLS SCO.
With both OMPS instruments onboard the same NPP satellite, the time difference between the
limb and nadir measurements is about 7 minutes (similar to Aura MLS and OMI instruments).
The OMPS data including evaluation of data quality are available from
https://ozoneaq.gsfc.nasa.gov/data/omps/.

All satellite-derived TCO represents monthly-means under mostly clear-sky conditions with
radiative cloud fractions < 40%. This cloud threshold reduces the number of total column ozone
pixels by ~20%. The cloud filtering was applied to reduce precision error in satellite-measured
TCO due to errors in assumed climatological below-cloud ozone for thick cloud scenes. These
errors in tropospheric ozone are largely random in nature on a pixel-by-pixel basis and do not
affect calculated trend magnitudes whether or not such measurements are removed from the
analyses. Satellite-derived TCO was gridded to $5^o \times 5^o$ bins centered on longitudes -177.5°, -
172.5°, …, 177.5°, and latitudes -27.5°, -22.5°, …, 27.5° for TOMS and latitudes -57.5°, -52.5°,
…, 57.5° for OMI/MLS (and also OMPS). This bin size for all measurements was chosen for
consistency because the original bin size for the CCD measurements for 1979-2005 is $5^o \times 5^o$.

2.2. MERRA-2 GMI Model.
The Modern-Era Retrospective analysis for Research and Applications (MERRA-2) GMI
simulation is produced with the Goddard Earth Observing System (GEOS) modeling framework
(*Molod et al*., 2015), using winds, temperature, and pressure from the MERRA-2 reanalysis
(*Gelaro et al*., 2017). The configuration for this study is a dynamically constrained replay (*Orbe
et al*., 2017) coupled to the Global Modeling Initiative's (GMI) stratospheric and tropospheric
chemical mechanism (*Duncan et al*., 2007; *Oman et al.,* 2013; *Nielsen et al*., 2017). The
simulation was run at ~0.5° horizontal resolution, c180 on the cubed sphere, and output on the
same 0.625° longitude x 0.5° latitude grid as MERRA-2 from 1980-2016.



The MERRA-2 GMI simulation includes emissions of NO, CO, and other non-methane
hydrocarbons from fossil fuel and biofuel sources, biomass burning, and biogenic sources.  There
are also NO emissions from lightning and soil.  Fossil fuel and biofuel sources are prescribed
from the MACCity Measuring Atmospheric Composition and Climate megaCity – zoom for the
environment (MACCity) inventory (*Granier et al*, 2011), which interpolates to each year from
the decadal Atmospheric Chemistry and Climate - Model Inter-comparison Project (ACCMIP)
emissions (*Lamarque et al*, 2010) and applies a seasonal scaling factor.  The MACCity inventory
ends in 2010, so for later years we use fossil fuel and biofuel emissions from the Representative
Concentration Pathways 8.5 (RCP8.5) scenario.  Time-dependent biomass burning emissions for
1997 onwards come from the Global Fire Emissions Dataset (GFED) version 4s (*Giglio et al.*,
2013).  Biomass burning emissions for prior years have interannual variability from regional
scaling factors based on the TOMS aerosol index (*Duncan et al*, 2003) imposed on a climatology
derived from GFED-4s, similar to the approach used in *Strode et al.* [2015].   Emissions of
isoprene and other biogenic compounds are calculated online using the Model of Emissions of
Gases and Aerosols from Nature (MEGAN) model [*Guenther et al.*, 1999, 2000], and thus
respond to MERRA-2 GMI meteorology.  NO emissions from soil, parameterized based on
*Yienger and Levy* [1995], also responds to the MERRA-2 meteorology. Lightning NO production
is prescribed monthly based on the scheme described in *Allen et al*. (2010) using a de-trended
cumulative mass flux in the mid-troposphere from MERRA-2, constrained seasonally with the
OTDLIS v2.3 climatology (*Cecil et al.*, 2014).  TCO is derived from the GMI simulation by
integrating the generated ozone profiles from the surface up to tropopause pressure.  GMI TCO
(discussed below) was also averaged monthly and re-gridded from original 0.5º latitude × 0.625º
longitude resolution to this same 5º × 5º gridding.  Where we refer to GMI in this paper it is
equivalent to MERRA-2 GMI.

2.3. Ozonesondes.

We include balloon-launched ozonesonde measurements for comparisons and validation of the
OMI/MLS TCO.  The ozonesonde database extends from 2004-2016 and includes measurements
from Southern Hemisphere ADditional OZonesondes (SHADOZ) (Thompson et al., 2017; Witte
et al., 2017), World Ozone and Ultraviolet Data Center (WOUDC) (https://woudc.org/), and



Network for the Detection of Atmospheric Composition Change (NDACC).
(http://www.ndsc.ncep.noaa.gov/). The ozonesondes provide daily ozone profile concentrations
as a function of altitude from several dozen global station sites. The ozone profiles are
integrated vertically each day to derive tropospheric column measurements. Most all of the
sonde ozone profile measurements are derived from Electrochemical Concentration Cell (ECC)
instruments. The Supporting Material section discusses the ozonesonde analyses that include
evaluation of potential offset and/or drift in OMI/MLS data.

2.4. Trend Calculations.

TCO offset differences between TOMS and OMI/MLS measurements are found to be regionally
varying with offset difference values up to 5 DU or greater which hampers any useful effort for
deriving trends from their concatenated datasets. Offsets of several DU between TOMS and
OMI total ozone have been well documented (e.g., Witte et al., 2018, and references therein).
Therefore, we have calculated trends independently for the TOMS (1979-2005) and OMI/MLS
(2005-2016) datasets. Total net change in TCO (in DU) at each grid point for the 38-year record
was determined by adding together the net changes (i.e. trend in DU-month$^{-1}$ × number of
months) for the TOMS and OMI/MLS records. Year 2017 and later months were not included in
our analyses because the MERRA-2 GMI simulation ended after 2016 and also that the global
ozonesonde measurements used for validating the OMI/MLS TCO extended only into mid-2016.

Multivariate linear regression (MLR) (Ziemke et al., 1997, and references therein) was applied to
estimate trends in TCO. The regression includes components for the seasonal cycle, linear trend,
and ENSO (e.g., Nino 3.4 index) from $TCO(x,t) = A(x,t) + B(x,t) \cdot t + C(x,t) \cdot Nino3.4(t) + \varepsilon(x,t)$,
where $x$ is the grid point and $t$ is month. The term $\varepsilon(x,t)$ represents residual error. We applied
two approaches regarding $Nino3.4(t)$ in the MLR model. One approach was to de-trend
$Nino3.4(t)$ prior to the regression analysis and the other was not to de-trend this proxy. A main
reason for possibly wanting to de-trend $Nino3.4(t)$ is that TCO variability is not truly linear with
$Nino3.4(t)$ variability over any timescale including decadal which may potentially influence
linear trend calculations in the MLR method. We opted not to include de-trending of $Nino3.4(t)$
after finding little or no difference between either approach for both OMI/MLS and TOMS



records. The seasonal coefficient *A* in the MLR equation above includes a constant plus annual
and semi-annual harmonics while coefficients *B* and *C* each include a constant. Since our study
does not evaluate seasonality of trends, we constrained the number of regression constants for
trend *B* to only one which tends to improve overall trend statistical uncertainties when compared
to using several regression seasonal constants for *B*. Trend magnitudes exceeding the calculated
2σ value uncertainty for *B* are deemed statistically significant. Calculated 2σ uncertainties for
trends included an autoregressive-1 adjustment as presented in Weatherhead et al. (1998).
Trends were calculated similarly for GMI TCO and NO emissions using this MLR approach.

**3. Trends in Tropospheric Ozone.**

3.1. The Aura Record (2005-2016).

OMI/MLS TCO trends for 60ºS - 60ºN are shown in Figure 1a with asterisks denoting regions
that are statistically significant at 2σ level. Positive trends lie in the tropics and extra-tropics in
both hemispheres with the largest trends (shown in red) of ~+3 DU-decade$^{-1}$ or greater extending
from India to East/SE Asia and further eastward over the Pacific Ocean. There are also
statistically significant increases in ozone in the north Atlantic and Africa.

Trends for GMI TCO (Figure 1b) have several features similar to trends for OMI/MLS TCO.
Large positive trends for GMI also extend from Saudi Arabia and India to SE/East Asia and
further eastward over the Pacific Ocean. Changes for both OMI/MLS and GMI TCO over this
region are ~+3 DU-decade$^{-1}$. GMI and OMI/MLS TCO also indicate positive trends extending
from the tropical/subtropical Atlantic to Africa. There are clear differences between GMI and
OMI/MLS in Figure 1, such as in the SH where GMI does not indicate statistically significant
positive trends as the satellite observations do. Anet et al. (2017) examined surface ozone data
from El Tololo, Chile (30ºS, 71ºW) and found a small positive trend of ~+0.7 ppbv-decade$^{-1}$ for
the period 1995-2010. Their analyses indicated that the positive increase at the site was driven
mainly by stratospheric intrusions and not photochemical production from anthropogenic and
biogenic precursors. The results from Anet et al. (2017) suggest that the positive trends in SH



OMI/MLS TCO in Figure 1a (primarily over ocean) may be real; however, one cannot make any
conclusion based on only ground-level measurements and from only one station.

We have calculated ozonesonde column ozone trends for the same 2005-2016 Aura record to
compare with the GMI and OMI/MLS TCO trends in Figure 1.  (The Supporting Material
discusses these trend comparisons.)  Figure S10 of the Supporting Material indicates that it is not
possible from the ozonesondes to conclude anything definitive regarding trends, particularly in
the SH extra-tropics where the ozonesondes are relatively scarce over the short Aura time record.

Trends for NO emissions for 2005-2016 from the GMI simulation are shown in Figure 2, again
with positive (negative) trends as red (blue).  Largest increases in tropospheric NO in Figure 2
are located over India and east/SE Asia while greatest decreases originate over the eastern US,
Europe, and Japan.  We note that although there are large increases in NO emissions over eastern
China for 2005-2016 depicted in Figure 2, observations show $NO_2$ decreased over this region
after year 2012 (e.g., Krotkov et al., 2016).  This recent downturn is not included in the GMI
emissions, likely contributing to the overestimate of the ozone trend over eastern China in the
GMI simulation.  Overall, however, the ability of the GMI simulation to capture the positive
trends above and downwind of regions with large $NO_x$ emission increases suggests that the $NO_x$
emission trends are driving the trends in TCO over India and east Asia.

Figure 1 shows that the regions of large decrease in NO such as the eastern US and Europe in
Figure 2 do not coincide with similar decrease in TCO for either GMI or OMI/MLS.  Both GMI
and OMI/MLS TCO instead show essentially zero or slightly positive trends for these regions,
despite the fact that the GMI simulation indicates significant negative trends in tropospheric
column $NO_2$ over the eastern U.S. and Europe.  This contrasts with the situation at the surface, in
which simulations with GMI chemistry indicate decreases in surface ozone over the eastern U.S.
in response to $NO_x$ reductions (Strode et al., 2015).

Figure 3 shows comparisons between OMI/MLS and GMI deseasonalized TCO time series and
their calculated linear trends for (a) SE Asia, (b) equatorial Africa, (c) NE Pacific, and (d) north
Atlantic.  Included in each panel are MLR regression fits for linear trends and their calculated 2σ



uncertainties (both in DU-decade$^{-1}$). Not only are trends for GMI and OMI/MLS comparable and
statistically significant in Figure 3 in each panel, but their month-to-month variations in their de-
trended time series have relatively large cross-correlations varying from +0.64 to +0.70. Several
inter-annual features are common with both MERRA-2 GMI and OMI/MLS TCO time series in
Figure 3 such as large reductions (exceeding -5 DU) during spring 2008 over the NE Pacific and
spring 2010 in the north Atlantic.

3.2. The TOMS Record (1979-2005).

Trends for TOMS (1979-2005) and GMI (1980-2005) TCO are shown in Figure 4. As with both
OMI/MLS and GMI TCO for the Aura period 2005-2016 in Figure 1, largest positive trends in
Figure 4 are also located over the Near East to East Asia and extending further eastward over the
Pacific Ocean. Calculated trends for this region are ~+1.2 to +1.4 DU-decade$^{-1}$ for both TOMS
and GMI which are considerably smaller than during the Aura record. An important conclusion
is that both the model and measurements in Figures 1 and 4 suggest that the trends in
tropospheric ozone over this region are markedly larger during the Aura period compared to the
earlier TOMS period, by a factor of about 2-2.5.

As with OMI/MLS and GMI TCO trends in Figure 1 there are discrepancies between the TOMS
and model TCO trends in Figure 4. For TOMS TCO in Figure 4 there are regions of negative
trends (in blue) as much as -0.6 DU-decade$^{-1}$ over ocean in both hemispheres that are not
explainable. Trends for GMI in Figure 4 are instead largely positive within these regions and
actually positive throughout much of the SH when compared with TOMS. This suggests that the
TOMS trends may be biased slightly low overall, provided that the simulation is closer to truth.

The trends for GMI TCO are positive over Brazil whereas OMI/MLS TCO shows only a hint of
positive trends. It is likely that there will be smaller trends for TOMS because most ozone
produced from biomass burning over Brazil lies in the low troposphere, and also that TOMS has
reduced ability to detect ozone in the low troposphere. The GMI simulation shows that of the
~+1.4 DU-decade$^{-1}$ TCO trend over Brazil in Figure 4, about +0.9 DU-decade$^{-1}$ of this trend
comes from ozone in the low troposphere below 500 hPa. With a known retrieval efficiency of



50-60% below 500 hPa (and essentially 100% above 500 hPa) for TOMS over Brazil, the model
suggests that TOMS should detect a trend of about +0.5 DU-decade$^{-1}$ below 500 hPa.  Therefore
TOMS would then have a trend in TCO of about +0.9 DU-decade$^{-1}$ which is comparable to the
~+0.8 DU-decade$^{-1}$ measured for TOMS in Figure 4.

In Figure 6 we show some examples of time series of TCO for TOMS and MERRA-2 GMI in
regions where both records exhibit statistically significant positive trends.   The positive
correlations between TOMS and model TCO in Figure 6 are generally small compared to the
correlations between OMI/MLS and model TCO in Figure 3.   The only large correlation in
Figure 6 is over Indonesia and is due to the intense El Nino of 1997-1998 that caused record
increases in TCO in October 1997 in the region due to record levels of biomass burning (e.g.,
Chandra et al., 2003).   The cross-correlations in the other panels in Figure 6 are small; these
smaller correlations indicate the noisy nature of TOMS measurements compared to OMI/MLS
and also possibly larger uncertainties present in meteorological winds, temperatures, and
emissions during these earlier TOMS years for the GMI simulation.

A main result from Figures 4 and 6 is that the positive trends for both TOMS and MERRA-2
GMI TCO are substantially larger, by a factor of about 2 or more, during the OMI/MLS record
compared to the TOMS record.  The GMI simulation suggests that larger trends during the Aura
record are the manifestation of an escalation of anthropogenic emissions and transport.

3.3. The Merged Record (1979-2016).

The net increases in tropospheric ozone over India and east/SE Asia for the merged 38-year
record are sizable.  Total changes in GMI and satellite-measured TCO for the merged record are
shown in Figure 7 where contour values were determined by adding changes from the individual
TOMS and OMI/MLS records together.  There are two regions of greatest increase of TCO in
Figure 7 for both GMI and the satellite measurements, one coinciding with the Near East to East
Asia (increases of ~+6 to +7 DU, or about 15-20% average background ozone) and the other
being tropical Africa/Atlantic (increases of ~+4 to +5 DU, of about 10-15% average background
ozone).  There is also an area of negative net change in the SH lying between Australia and the





maritime continent in Figure 7 for both GMI and measurements (shown in blue); these negative
variations over the SH Indian Ocean appear small and are not statistically significant.

The color bar in Figure 7 also provides conversion from DU to tropospheric ozone mass surface
density in units of metric tons per km$^2$. This conversion was included primarily to compare our
results with the model simulation of Zhang et al. (2016). The large TCO trends over India and
east/SE Asia in Figure 7 are about +0.13 to +0.15 metric tons per km$^2$ for both GMI and the
satellite data. These numbers are comparable to increases of ~+0.11 metric tons per km$^2$ for this
region as modeled by Zhang et al. (2016) for years 1980-2010.

Figure 8 shows TCO time series from the merged satellite measurements for 1979-2016 centered
over the two regions of largest increase in Figure 7 (i.e., eastern Asia and equatorial Africa). In
both panels TOMS is the solid red curve and OMI/MLS is the dotted blue curve. For plotting
purposes, offsets were applied to the TOMS data in both panels using 2005 overlap
measurements (see figure and caption). The last five years in both panels in Figure 8 shows that
current OMPS TCO (solid black curves) with several years of overlap with OMI/MLS TCO will
be useful to continue the OMI/MLS record which has already extended past 13 years.

Studies suggest that ozone in the lower stratosphere in both hemispheres has been decreasing
over the last 1-2 decades despite the decrease in global CFCs following the 1987 Montreal
Protocol. Ball et al. (2018) evaluated global ozone trends for 1985-2016 by combining models
with measurements from several satellite instruments. A conjecture as stated by Ball et al.
(2018) is that while ozone in the upper stratosphere above ~10 hPa appears to be recovering,
ozone in the lower stratosphere appears to be decreasing which models do not seem to replicate
despite the decrease in CFCs. A main point of Ball et al. (2018) is that total ozone has not
changed because the ongoing stratospheric ozone decrease is opposed by tropospheric ozone
increase. A global decrease in lower stratospheric ozone of about 2 DU below 32 hPa was
detected by Ball et al. (2018) and it appeared to be compensated largely by opposite increases in
tropospheric ozone. In their study they included OMI/MLS TCO for 2005-2016 (i.e., their
Figure 4 and Figure S13) and measured a trend in 60$^{\circ}$S-60$^{\circ}$N TCO of about +1.7 DU-decade$^{-1}$
which mostly cancels out the negative trend in stratospheric ozone. Wargan et al. (2018) in a





related paper evaluated MERRA-2 assimilated ozone for 1998-2016 using an idealized
atmospheric tracer also driven from MERRA-2 meteorological fields. Similar to Ball et al.
(2018), Wargan et al. (2018) also found net decrease in ozone in the lower stratosphere (i.e.,
within a 10 km layer above the tropopause) in both hemispheres; their trend values were about -
1.2 DU-decade$^{-1}$ in the SH and about -1.7 DU-decade$^{-1}$ in the NH. Wargan et al. (2018) found
evidence that these negative trends over the last two decades have been driven by enhanced
isentropic transport of ozone between the tropical and extratropical lower stratosphere.

The increases in measured TCO from TOMS and OMI/MLS as indicated in Figures 1, 3, 4 and
in Figures 6-8 can have implications for evaluating global ozone trends, particularly for trends in
total column ozone and assessment of the recovery of stratospheric ozone. One should be careful
using total ozone to infer stratosphere ozone recovery if trends in TCO are not accounted for.
The increases in TCO of +6 to +7 DU in Figures 7-8 for India-eastern Asia represent a sizeable
change even for total column ozone.

**4. Summary.**

Studies suggest that ozone in the troposphere has increased globally throughout much of the 20th
century due largely to increases in anthropogenic emissions. We provide evidence from
combined satellite measurements and a chemical transport model that tropospheric ozone over
the last four decades does indeed indicate increases that are global in nature, yet highly regional
due to combined effects of regional pollution and transport.

We have obtained tropospheric ozone trends for 1979-2016 by merging TOMS (1979-2005) and
Aura OMI/MLS (2005-2016) satellite measurements. We included the MERRA-2 GMI CTM
simulation to evaluate and possibly explain the global trend patterns found for both TOMS and
OMI/MLS TCO. Trends were calculated independently for TOMS and OMI/MLS records using
a linear regression model. Net changes in both measured and modeled TCO for the entire
merged record were estimated by adding net changes for the TOMS and OMI/MLS time periods
together.



A persistent trend pattern emerges with TCO for the GMI simulation and satellite measurements
for both the TOMS and OMI/MLS records.  The GMI model, and also measurements from
TOMS and OMI/MLS all independently show large (positive) trends in TCO in the NH
extending from the Near East to India and east/SE Asia, and further eastward over the Pacific
Ocean.  An important finding is that the trends in TCO for both the GMI model and satellite
measurements for this region are smaller during the earlier part of the merged record; that is, the
trends for both GMI and satellite measurements increase from about +1.2 to +1.4 DU-decade$^{-1}$
(1979-2005) to about +3 DU-decade$^{-1}$ or greater (2005-2016).  Analysis of the NO emissions
input to the GMI simulation indicates that the measured trends in tropospheric ozone in this
region including the escalation of increased trends during the latter Aura period are consistent
with increases in pollution in the region.

For the long merged record there are again strong similarities between the GMI simulation and
satellite measurements of TCO.  Net changes in tropospheric ozone for India and east/SE Asia
for 1979-2016 are about +6 to +7 DU, or about 0.13-0.15 metric tons per km$^2$ for both the GMI
and satellite TCO.  These are pronounced increases in TCO representing ~15-20% average TCO
background amounts.  Both the GMI simulation and satellite measurements show that of these
+6 to +7 DU increases over this broad area, about half or slightly most of the change (i.e., ~+4
DU) occurs during the Aura time record of 2005-2016.  The GMI simulation and satellite
measurements also depict a secondary maximum of TCO increase for 1979-2016 over the
tropical Atlantic/Africa region of about +4 to +5 DU (~10-15% average background ozone).

**Acknowledgments.** We thank the NASA Goddard Space Flight Center Ozone Processing Team
for the TOMS and OMI total ozone measurements and the Jet Propulsion Laboratory MLS team
for MLS v4.2 ozone.  OMI is a Dutch-Finnish contribution to the Aura mission.  We also thank
WOUDC and the NDACC for providing extensive ozonesonde measurements that we used for
the comparisons/validation of satellite tropospheric ozone.   Funding for this research was
provided in part by NASA NNH14ZDA001N-DSCOVR.   We also thank the NASA MAP
program for supporting the MERRA-2 GMI simulation and the NASA Center for Climate
Simulation (NCCS) for providing high-performance computing resources.  More information on
the    MERRA-2    GMI    simulation    and    access    is    available    at    https://acd-




ext.gsfc.nasa.gov/Projects/GEOSCCM/MERRA2GMI/. Tropospheric ozone data used in this
study are available from NASA Goddard Space Flight Center at http://acdb-
ext.gsfc.nasa.gov/Data_services/cloud_slice/ and links from the Aura Validation Data Center
(https://avdc.gsfc.nasa.gov/).

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









**FIGURES AND FIGURE CAPTIONS**



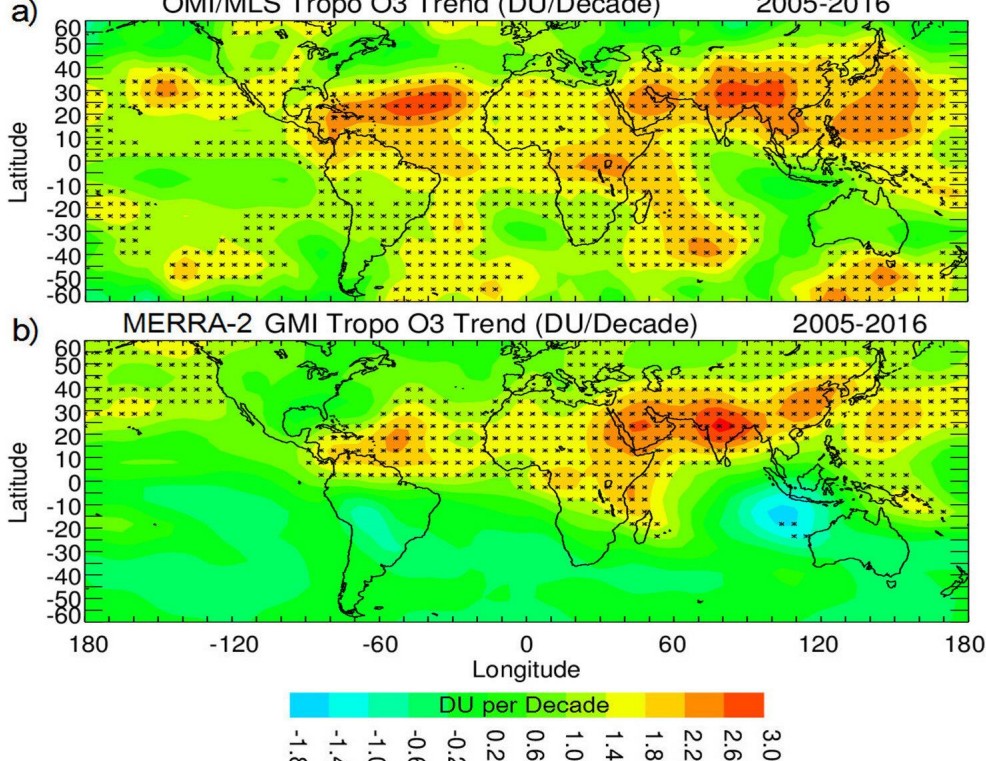


**Figure 1.** (a) Trends in OMI/MLS TCO (in DU-decade$^{-1}$) for 2005-2016. Asterisks denote grid

points where trends are statistically significant at the 2σ level. (b) Same as (a) except for

MERRA-2 GMI TCO.







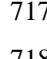

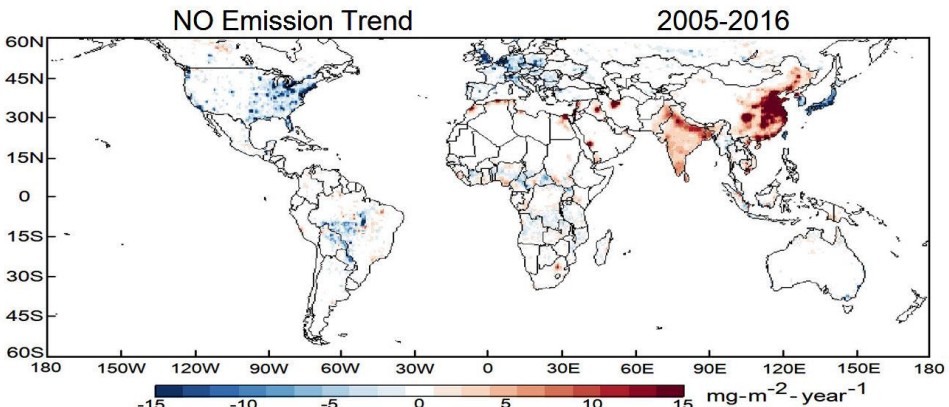


**Figure 2.** Trends in MERRA-2 GMI NO emissions (units mg-m$^{-2}$-y$^{-1}$) for 2005-2016.



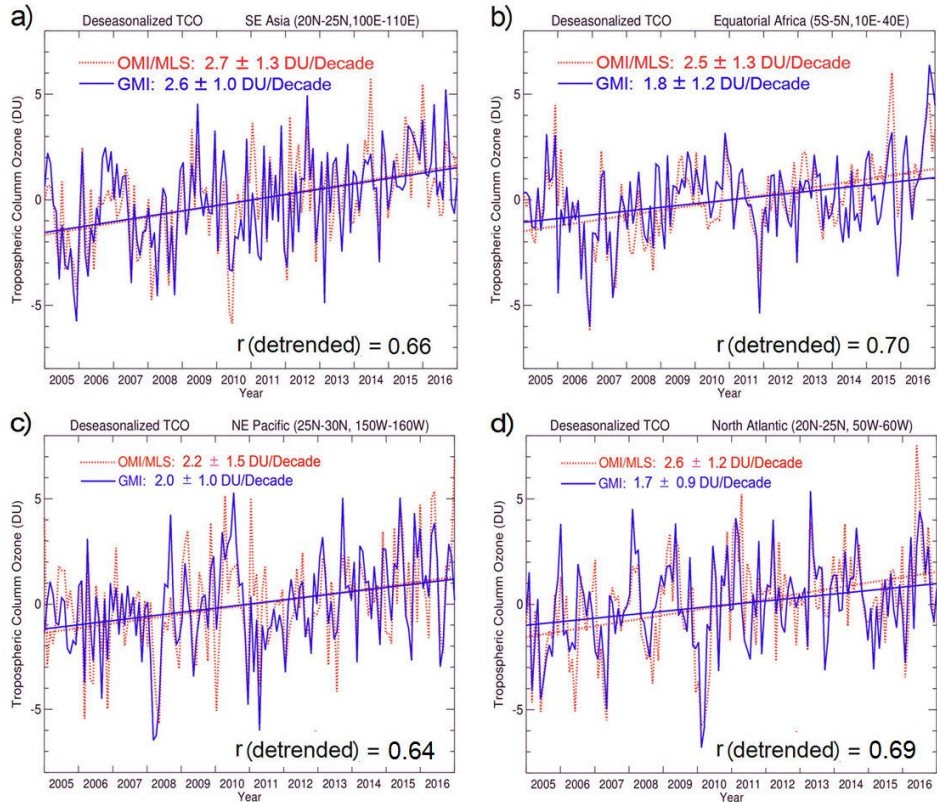





**Figure 3.** (a) Deseasonalized TCO for OMI/MLS (red, dashed curve) and the MERRA-2 GMI
model (blue, solid curve) for SE Asia. Included are MLR regression fits for linear trends and
calculated 2σ values (both in DU-decade$^{-1}$). Shown at the bottom is the correlation r between the
two time series after removing their linear trends. (b) Same as (a), but for equatorial Africa. (c)
Same, but for NE Pacific. (d) Same, but for north Atlantic.



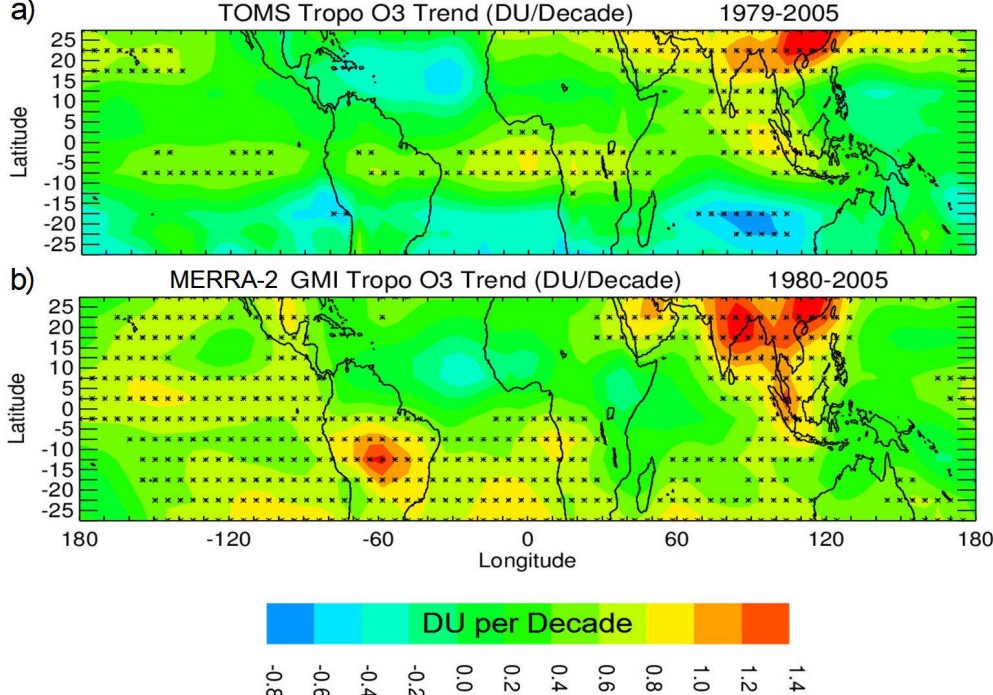


**Figure 4.** (top) Trends (DU-decade$^{-1}$) calculated for TOMS CCD TCO measurements for years
1979-2005. Asterisks denote grid points where trends are statistically significant at the 2σ level.
(bottom) Similar to (top), but for MERRA-2 GMI TCO and for 1980-2005.





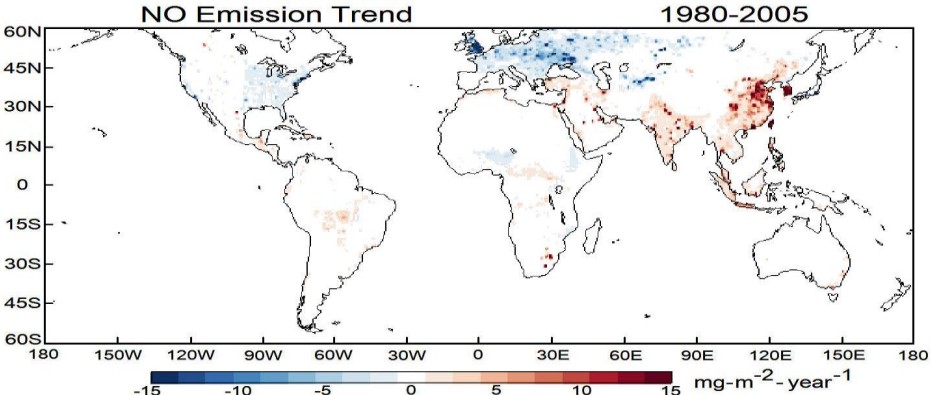


**Figure 5.** Trends in MERRA-2 GMI NO emissions (units mg-m$^{-2}$-y$^{-1}$) from biomass burning

for 1980-2005.


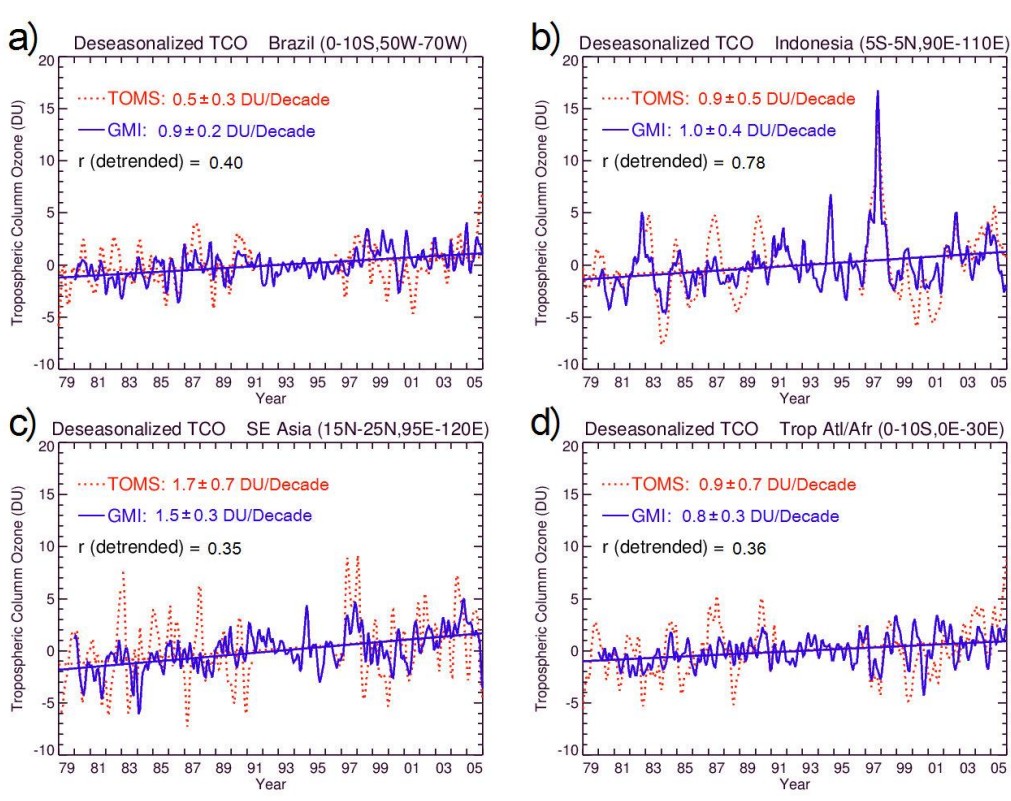


**Figure 6.**  (a) Deseasonalized TCO for TOMS (red, dashed curve) and the MERRA-2 GMI
model (blue, solid curve) for Brazil.  Included are their MLR linear trends and calculated $2\sigma$
values (both in DU-decade$^{-1}$) averaged over the specified region.  Shown also is the cross-
correlation r between the two time series after removing their linear trends.  (b) Same as (a), but
for Indonesia.  (c) Same as (a) but for SE Asia.  (d) Same as (a) but for tropical Atlantic/Africa.


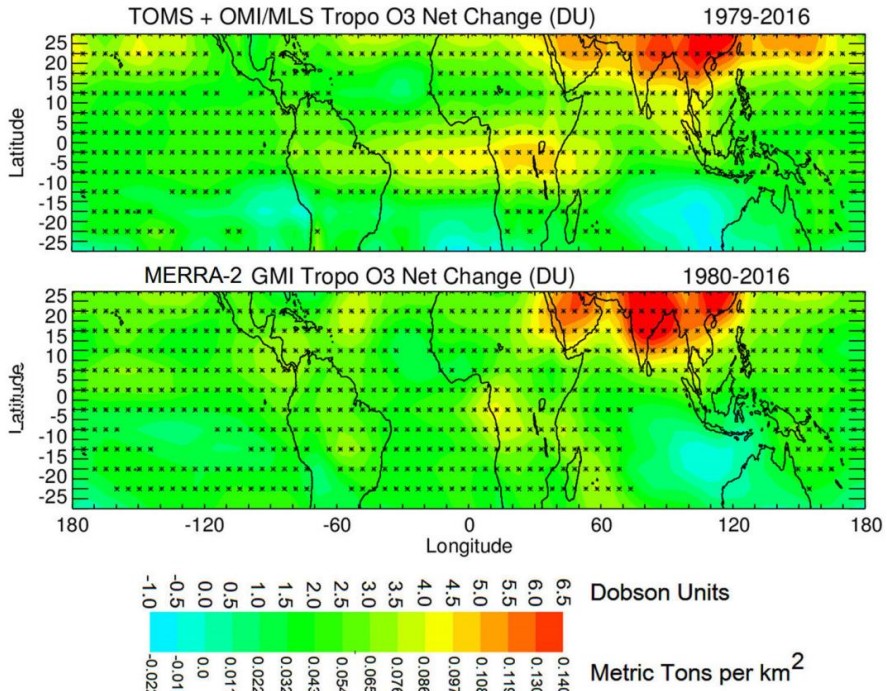



**Figure 7.**  (top) Net changes in TOMS and OMI/MLS TCO calculated for their combined time
records (1979-2016).  The net changes for TCO are shown in the color bar in both DU and
metric tons of ozone per km$^2$ (1 DU ≡ 0.0214 metric tons per km$^2$ for ozone).  Asterisks denote
grid points where net changes are statistically significant at the $2\sigma$ noise level.  (bottom) Similar
to (top), but for GMI TCO and years 1980-2016.  Net change for GMI TCO is determined
similar to the satellite measurements by adding together the net changes for the two records (i.e.,
for GMI, the 1980-2005 and 2005-2016 periods).






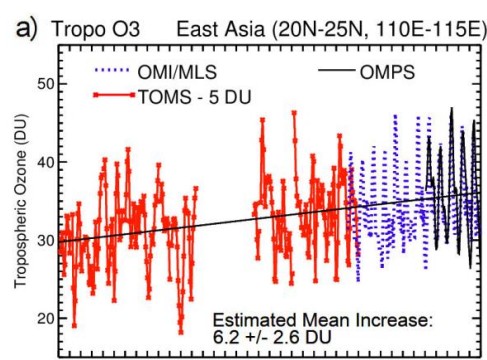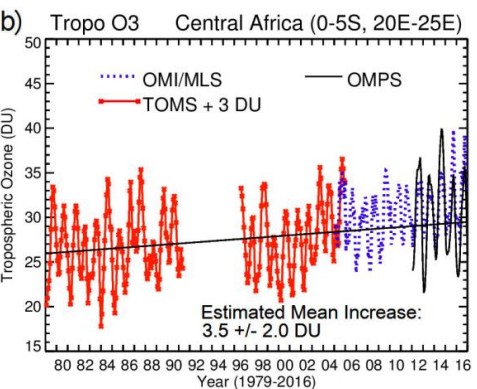



**Figure 8.** (a) Merged time series of TOMS/OMI/MLS/OMPS TCO for 1979-2016 over east
Asia centered at 22.5° N and 112.5° E (5° × 5° region). The solid red curve is TOMS TCO and
dashed blue curve is OMI/MLS. OMPS TCO (solid black curve) is also over-plotted with
OMI/MLS TCO starting 2012 for comparison. A constant adjustment of about -5 DU (using
year 2005 coincident overlap data) was applied to the TOMS measurements for plotting with
OMI/MLS. Both OMI/MLS and OMPS TCO also included offsets of +2 DU and -2 DU
following comparisons with ozonesonde measurements (see Supplementary Material). The
indicated total increase of 6.2 DU was estimated using a regression best-fit line (black line
shown) to the TOMS/OMI/MLS merged time series and agrees well with the 6-7 DU net
increase for this region in Figure 7. (b) Similar to (a) except for central Africa centered at 2.5° S,
22.5° E and a TOMS offset of +3 DU. The line-fit increase is slightly smaller than the 4-5 DU in
Figure 7. The estimated mean increases in both panels include calculated $2\sigma$ uncertainties.
