# Peer review of "Trends in Global Tropospheric Ozone Inferred from a Composite Record of TOMS/OMI/MLS/OMPS Satellite Measurements and the MERRA-2 GMI Simulation"

_Atmospheric Chemistry and Physics, 2018_

## Referee Comment (RC1) · Anonymous Referee #1 · 20 Dec 2018

Main Comments: You need to improve the connection between the main article and supporting material. Specifically, you need to refer to sections A, B, C, and D of the supporting material separately and to restate (in one or two sentences) the main conclusion of these sections in the main bod (e.g.., the magnitude of the various constant offsets)

More information on the likely cause(s) of the increase in tropospheric ozone column over central Africa would be useful.

[Figure]

Comments:

L66: How different is a 28 Tg from what you find? If significantly, different, the cause could be discussed around lines 372-377.

L126: Remind readers why the CCD product is limited to the tropics

L350-352: What drifts have been observed in the MERRA-2 meteorological fields during the TOMS period that might affect the trends in the GMI simulation?

Minor Comments:

L28: is include to evaluate –> is used to aid in the interpretation of

L60: effects on tropospheric ozone from these changes in emissions –> effects of these changes in emissions on tropospheric ozone

L92: was determined –> was constructed

L203: v2.3 climatology –> v2.3 lightning climatology

L220: that include –> and includes

L286: tropospheric NO –> tropospheric NO emissions

Supporting Material L15: Remind reader why you use only rows 3-18 here.

L23: Figure S1. What do you mean by "Overkill" TCO?

L23: You may want to include the mean trend by decade for each region as these trends were used as a guide when choosing -1.0 DU decade-1 as the OMI/MLS TCO adjustment.

L29: (indicated) –> beginning with 40N-60N (upper left) and ending with 40S-60S (lower right).

L64: Likely fine but confirm that change and uncertainty are identical.

L98-103: Make sure that this information is in main paper too.

L107: An important yet small –> A small but important

L107: is to show some –> is an

L167: "Most all". Can you be more specific?

L199: Why did you integrate from the ground to 8km as opposed to from the ground to the thermal tropopause as done elsewhere in the article?

Figure S10: The captions for A and B are identical. I believe the caption for B should refer to GMI as opposed to OMI/MLS.

---

## Referee Comment (RC2) · Anonymous Referee #2 · 11 Jan 2019

General Comments

This paper presents satellite tropospheric column ozone measurements from various instruments over the period 1979-2016, and compares with a simulation from a chemistry transport model driven with reanalysis meteorology and best estimates of changing emissions. The comparison indicates similar upwards ozone trends with a similar regional time evolution (accelerating increases over Asia in recent years). The data and model comparison are well presented and the overall story is convincing, and well worthy of publication. The model simulation details should be expanded a little (see

below), to stress that the ozone increases are not simply due to increases in NO emissions. Indeed, it would be most interesting to extend the modelling work to more fully understand the drivers of the ozone increase (e.g. the components from methane, NOx, any changes in stratospheric contribution, any changes in ozone lifetime, e.g. due to changes in deposition or humidity), although I can imagine the authors will say this is beyond the scope of the current publication. Nevertheless, if they can say anything about attribution that would be most useful, especially from a policy perspective – we would like to understand the processes that have led to the increases in ozone seen, in order to reduce/reverse them in future. In a few places, the paper lapses into overly technical jargon, but on the whole it is clear and well written. If these points can be clarified, and the specific points below addressed, I fully recommend this paper for publication.

Specific Comments

L30 The GMI simulation is definitely not 'identical' to the satellite measurements. (It would be worrying if it was.)

L31 Define TCO.

L38 N Atlantic

L42 ...changes in emissions and concentrations of global pollutants,...

L46 The Lin et al. (2017) study appears to focus on the US rather than being a truly global study (cf. Young et al., 2013, for example). [Young, P.J. et al. (2013) doi:10.5194/acp-13-2063-2013]

L51 The Shepherd et al. (2014) paper is mainly about stratospheric, rather than tropospheric ozone, so also seems an odd choice at this point.

L120 [I complained about this in my initial report, but it's still here!] "...developed within NASA Goddard Code 614..." I don't know what this means. Is it a building or an institute? Is it a protocol or some sort of NASA standard method that we are all

supposed to know? It is technical jargon that should be decoded for the non-NASA general public readership. At least give us a reference.

L143 What are "...in situ UV cloud pressures..."?

L201 What is a "...de-trended cumulative mass flux..."?

L208 The model description doesn't mention several important aspects for ozone. How is methane handled? (The ozone trends will have been partly driven by methane trends, but it is not mentioned at all). The focus is on emissions – but what about ozone removal? What does the ozone deposition scheme look like? Is it related to land cover properties (e.g. Leaf Area Index, etc.), and does this change? How does effective stratospheric Cl loading vary?

L218 Most all?

L225 What are 'TCO offset differences'? They are not explained. Is there an overlapping period of both TOMS and OMI/MLS in 2005?

L233 'after 2016' is not very specific.

L286 tropospheric NO emissions...

L289 NO2 concentrations? Please clarify that emissions are not equivalent to, or to be used interchangeably with, concentrations. This is fundamentally important.

L296 NO emissions...

L334, 335, 337 lower

L343 I don't think Figure 5 is referred to (either at all, or before Figure 6).

L388 CFC concentrations?

L448 most -> more

L725 Does Figure 5 (which, as noted above, is not referred to in the main text) really

show trends in biomass burning emissions, as the caption indicates?

---

## Author Comment (AC1) · 15 Feb 2019

Main Comments: You need to improve the connection between the main article and supporting material. Specifically, you need to refer to sections A, B, C, and D of the supporting material separately and to restate (in one or two sentences) the main conclusion of these sections in the main bod (e.g.., the magnitude of the various constant offsets

Thanks – this is an important point. In the revision we mention each section A-D

individually in the main text including their main conclusion(s).

More information on the likely cause(s) of the increase in tropospheric ozone column over central Africa would be useful.

We added further discussion of the Heue et al. (2016) results that indicated increases in biomass burning as the likely cause of positive trends over that region. Their analysis suggested that positive trends in ozone over central Africa maximized for the months of June-August which coincides with the peak burning season in that region.

Comments:

L66: How different is a 28 Tg from what you find? If significantly, different, the cause could be discussed around lines 372-377.

We have added more discussion in the revision.

L126: Remind readers why the CCD product is limited to the tropics

Done.

L350-352: What drifts have been observed in the MERRA-2 meteorological fields during the TOMS period that might affect the trends in the GMI simulation?

We have added discussion on this point regarding changes in the observing system input to MERRA-2 and impact on long record ozone.

Minor Comments:

L28: is include to evaluate –> is used to aid in the interpretation of

Done.

L60: effects on tropospheric ozone from these changes in emissions –> effects of these changes in emissions on tropospheric ozone

L92: was determined –> was constructed

[Figure]

Done.

L203: v2.3 climatology –> v2.3 lightning climatology

Done.

L220: that include –> and includes

Done.

L286: tropospheric NO –> tropospheric NO emissions

Done.

Supporting Material L15: Remind reader why you use only rows 3-18 here.

Done.

L23: Figure S1. What do you mean by "Overkill" TCO?

Done.

L23: You may want to include the mean trend by decade for each region as these trends were used as a guide when choosing -1.0 DU decade-1 as the OMI/MLS TCO adjustment.

Done.

L29: (indicated) –> beginning with 40N-60N (upper left) and ending with 40S-60S (lower right).

Done.

L64: Likely fine but confirm that change and uncertainty are identical.

Done.

L98-103: Make sure that this information is in main paper too.

Added this also to main text.

L107: An important yet small –> A small but important

Done.

L107: is to show some –> is an

Done.

L167: "Most all". Can you be more specific?

Re-written / added text to clarify. Detailed in main text Section 2.3.

L199: Why did you integrate from the ground to 8km as opposed to from the ground to the thermal tropopause as done elsewhere in the article?

Sonde ground-to-8 km column ozone has now been replaced with sonde TCO for the analyses.

Figure S10: The captions for A and B are identical. I believe the caption for B should refer to GMI as opposed to OMI/MLS.

We have corrected this in the new Figure S10.

---

## Author Comment (AC2) · 15 Feb 2019

**General Comments**

This paper presents satellite tropospheric column ozone measurements from various instruments over the period 1979-2016, and compares with a simulation from a chemistry transport model driven with reanalysis meteorology and best estimates of changing emissions. The comparison indicates similar upwards ozone trends with a similar regional time evolution (accelerating increases over Asia in recent years). The data

and model comparison are well presented and the overall story is convincing, and well worthy of publication. The model simulation details should be expanded a little (see below), to stress that the ozone increases are not simply due to increases in NO emissions. Indeed, it would be most interesting to extend the modelling work to more fully understand the drivers of the ozone increase (e.g. the components from methane, NOx, any changes in stratospheric contribution, any changes in ozone lifetime, e.g. due to changes in deposition or humidity), although I can imagine the authors will say this is beyond the scope of the current publication. Nevertheless, if they can say anything about attribution that would be most useful, especially from a policy perspective – we would like to understand the processes that have led to the increases in ozone seen, in order to reduce/reverse them in future. In a few places, the paper lapses into overly technical jargon, but on the whole it is clear and well written. If these points can be clarified, and the specific points below addressed, I fully recommend this paper for publication.

Specific Comments

L30 The GMI simulation is definitely not 'identical' to the satellite measurements. (It would be worrying if it was.)

Done.

L31 Define TCO.

Done.

L38 N Atlantic

Done.

L42 ...changes in emissions and concentrations of global pollutants, ...

Done.

L46 The Lin et al. (2017) study appears to focus on the US rather than being a

truly global study (cf. Young et al., 2013, for example). [Young, P.J. et al. (2013) doi:10.5194/acp-13-2063-2013]

We added discussion of the Young er al. (2013) paper in this section in the revision.

L51 The Shepherd et al. (2014) paper is mainly about stratospheric, rather than tropospheric ozone, so also seems an odd choice at this point.

Shepherd et al. (2014) reference and discussion has been deleted in the revision.

L120 [I complained about this in my initial report, but it's still here!] "... developed within NASA Goddard Code 614 ..." I don't know what this means. Is it a building or an institute? Is it a protocol or some sort of NASA standard method that we are all supposed to know? It is technical jargon that should be decoded for the non-NASA general public readership. At least give us a reference.

This has been re-worded in the revision.

L143 What are "...in situ UV cloud pressures..."?

Re-worded.

L201 What is a "...de-trended cumulative mass flux ..."?

We have re-written this section to clarify with reference to the cumulative mass flux within the model.

L208 The model description doesn't mention several important aspects for ozone. How is methane handled? (The ozone trends will have been partly driven by methane trends, but it is not mentioned at all). The focus is on emissions – but what about ozone removal? What does the ozone deposition scheme look like? Is it related to land cover properties (e.g. Leaf Area Index, etc.), and does this change? How does effective stratospheric Cl loading vary?

These are good points which we have added discussion in the revision. We discuss

methane source and ozone deposition, etc. for the model.

L218 Most all?

We have added extensive discussion to clarify.

L225 What are 'TCO offset differences'? They are not explained. Is there an overlapping period of both TOMS and OMI/MLS in 2005?

Added text to clarify.

L233 'after 2016' is not very specific.

Re-worded to clarify.

L286 tropospheric NO emissions ...

Done.

L289 NO2 concentrations? Please clarify that emissions are not equivalent to, or to be used interchangeably with, concentrations. This is fundamentally important.

Re-worded now to clarify.

L296 NO emissions ...

Done.

L334, 335, 337 lower

Done.

L343 I don't think Figure 5 is referred to (either at all, or before Figure 6).

Thanks for seeing this. We didn't even discuss Figure 5 in the original draft. Figure 5 is now mentioned in the main text and has a corrected figure caption as well.

L388 CFC concentrations?

[Figure]

Done.

L448 most –> more

Done.

L725 Does Figure 5 (which, as noted above, is not referred to in the main text) really show trends in biomass burning emissions, as the caption indicates?

Very correct – the current figure is not just biomass burning NO, but all NO emissions just as in Figure 2.